# Patient-Oriented Perspective on Chemokine Receptor Expression and Function in Glioma

**DOI:** 10.3390/cancers14010130

**Published:** 2021-12-28

**Authors:** Damla Isci, Giulia D’Uonnolo, May Wantz, Bernard Rogister, Arnaud Lombard, Andy Chevigné, Martyna Szpakowska, Virginie Neirinckx

**Affiliations:** 1Laboratory of Nervous System Diseases and Therapy, GIGA Neuroscience, GIGA Institute, University of Liège, 4000 Liege, Belgium; Damla.Isci@uliege.be (D.I.); Bernard.Rogister@uliege.be (B.R.); alombard@chuliege.be (A.L.); 2Immuno-Pharmacology and Interactomics, Department of Infection and Immunity, Luxembourg Institute of Health, 1445 Strassen, Luxembourg; Giulia.Duonnolo@lih.lu (G.D.); May.Wantz@lih.lu (M.W.); Andy.Chevigne@lih.lu (A.C.); Martyna.Szpakowska@lih.lu (M.S.); 3Faculty of Science, Technology and Medicine, University of Luxembourg, 4365 Esch-sur-Alzette, Luxembourg; 4Neurology Department, University Hospital, University of Liège, 4000 Liege, Belgium; 5Neurosurgery Department, University Hospital, University of Liège, 4000 Liege, Belgium; 6Tumor Immunotherapy and Microenvironment, Department of Oncology, Luxembourg Institute of Health, 1445 Strassen, Luxembourg

**Keywords:** glioma, chemokine receptor, patient-derived transcriptomic data, malignant processes, tumor microenvironment

## Abstract

**Simple Summary:**

Chemokines and their receptors have been pointed out as key actors in a variety of human cancers, playing pivotal roles in multiples processes and pathways. The present study aims at deciphering the functions of several chemokine receptors in gliomas, starting from publicly available patient-derived transcriptomic data with support from the current literature in the field, and sheds light on the clinical relevance of chemokine receptors in targeted therapeutic approaches for glioma patients.

**Abstract:**

Gliomas are severe brain malignancies, with glioblastoma (GBM) being the most aggressive one. Despite continuous efforts for improvement of existing therapies, overall survival remains poor. Over the last years, the implication of chemokines and their receptors in GBM development and progression has become more evident. Recently, large amounts of clinical data have been made available, prompting us to investigate chemokine receptors in GBM from a still-unexplored patient-oriented perspective. This study aims to highlight and discuss the involvement of chemokine receptors—CCR1, CCR5, CCR6, CCR10, CX3CR1, CXCR2, CXCR4, ACKR1, ACKR2, and ACKR3—most abundantly expressed in glioma patients based on the analysis of publicly available clinical datasets. Given the strong intratumoral heterogeneity characterizing gliomas and especially GBM, receptor expression was investigated by glioma molecular groups, by brain region distribution, emphasizing tissue-specific receptor functions, and by cell type enrichment. Our study constitutes a clinically relevant and patient-oriented guide that recapitulates the expression profile and the complex roles of chemokine receptors within the highly diversified glioma landscape. Additionally, it strengthens the importance of patient-derived material for development and precise amelioration of chemokine receptor-targeting therapies.

## 1. Introduction

Gliomas are glial tumors of the central nervous system (CNS), which are categorized into different subtypes and clinical grades based on their histological features as well as molecular markers (according to the World Health Organization (WHO)) [1]. Adult-type diffuse gliomas represent the majority of primary brain tumors detected in adults, glioblastoma (GBM) being the most malignant subtype [1]. It accounts for 48.3% of malignant tumors of the adult central nervous system [2] and systematically results in fatal outcome for patients. For over 15 years, standard-of-care treatment has combined maximal safe surgical resection, radiotherapy and concurrent temozolomide-based chemotherapy [3]. Despite extensive preclinical and clinical research continuously aiming to improve therapeutic efficacy, GBM recurrence is commonplace and patient survival from the time of diagnosis remains low [4]. Several mechanisms underlie tumor relapse: (1) the infiltrative nature of GBM that invades and disseminates through the whole brain tissue [5]; (2) the multilevel heterogeneity of GBM tumors, which exhibit inter-patient and intra-tumoral disparities [6], include diverse cell types and cellular states [7]; and (3) the ability of GBM cells to interact with and adapt to their microenvironment [8], to interconnect with neighboring tumor cells [9] or to harness healthy brain cells [10]. These devious mechanisms together support GBM to escape and resist treatment.

Chemokines are a subfamily of chemotactic cytokines secreted by a wide range of cell types in various tissues and are important regulators of developmental processes, immune responses and tissue repair [11]. Chemokines exert their effect by activating G protein-coupled receptors, which triggers downstream signaling pathways leading to cell migration, modulation of gene expression and cell phenotypes [12,13]. They are classified into four subfamilies—CC, CXC, CX3C and XC—based on the arrangement of the cysteine motif in their N-terminal part, while their receptors are classified according to the type of chemokines they bind (CCR, CXCR, CX3CR and XCR). Recently, four chemokine receptors have been grouped in a subfamily of “atypical chemokine receptors” (ACKRs) owing to their inability to activate the classical ligand-induced G protein signaling cascades. They do however have an important regulatory role and can act as scavengers by reducing chemokine availability in the extracellular environment [14,15]. Chemokines and chemokine receptors have been proposed as key actors in cancer cell growth, migration, invasion, neovascularization, as well as in the fine-tuned interplay between tumor cells and tumor-associated immune cells [16,17]. The growing interest in chemokine receptor function in GBM is of complex nature. Not only are chemokines and chemokine receptors involved in GBM cell malignant phenotype, they also play an important part in the immune cell recruitment to the tumor. These molecules are therefore being increasingly considered as potential targets in immunotherapy approaches for GBM [18].

The last decade has witnessed an unprecedented effort in collecting samples and clinical data from patients suffering from solid cancers, including brain tumors. International consortia and multicenter projects (e.g., The Cancer Genome Atlas (TCGA) [19], Glioma Longitudinal AnalySiS (GLASS) consortium [20], Gliogene [21], etc.) have gathered considerable patient cohorts that provided the neuro-oncology community with large multi-omics datasets, offering invaluable information for the classification and grading of tumors as well as for the understanding of molecular mechanisms underlying glioma biology. Whereas multiple therapeutic strategies have thus far failed to translate from the bench to the clinic because of limited research tools, the availability of patient data and biological material now facilitates clinically relevant research and fosters the development of personalized therapies [22,23].

Here, we aim to refine the current knowledge about the role of chemokine receptors in glioma from a patient-oriented perspective. We analyzed publicly available datasets and highlighted a subset of receptors that appear to be significant in GBM patients, for which we gather and discuss recent insight from the literature.

The purpose of this study is to provide researchers in the field with a clinically-relevant, up-to-date practical resource that could orient the next steps toward chemokine receptor-based treatment for glioma patients. We voluntarily highlight the literature that describes data generated from patient material and mention preclinical data on cellular and animal models when considered pertinent. We do not detail the mechanistic and functional aspects of each described receptor, which were exhaustively reviewed recently [24].

## 2. Methods

We aimed to highlight putative variations in the expression of chemokine receptors in different types of gliomas, as well as in different tumor subregions and cellular subsets. To do so, we browsed four different glioma patient datasets using available online tools and exploited the data related to the information of interest (Table 1). We focused on gene expression data, generated by RNA sequencing of patient-derived residual tumor tissue, obtained after surgical resection. We analyzed the expression of 22 genes encoding for chemokine receptors, namely CC receptors 1 to 10 (CCR1-10), CX3C receptor 1 (CX3CR1), CXC receptors 1 to 6 (CXCR1-6), XC receptor 1 (XCR1) and the atypical chemokine receptors ACKR1 (or DARC), ACKR2 (or D6), ACKR3 (or CXCR7/RDC1) and ACKR4 (or CCRL1). The alternative gene names were used when required by the online platform. Original publications, online tools, RNA sequencing method as well as number of samples and patients included in the datasets are listed in Table 1. No recalculation nor modification of the existing data was performed. Figures included in this manuscript are either original heat maps displaying the unchanged data downloaded from the databases or were directly generated on the online platforms (for scRNAseq data).

## 3. Chemokine Receptor Expression in Gliomas

We first aimed to highlight which chemokine receptors are most abundantly expressed in gliomas. We unraveled the expression of 22 chemokine receptors in tumor tissue collected from newly diagnosed diffuse glioma patients using the TCGA LGG-GBM dataset (including 513 low-grade gliomas (LGG) and 154 GBM diagnosed patients with available RNAseq data). In an attempt to relate gene expression to glioma clinical subgroups associated with respective disease severity, we classified patients based on isocitrate deshydrogenase (IDH) mutation and 1p/19q co-deletion status, as these features were previously suggested to correlate with histological types and clinical grades. We therefore consider “IDH mutant 1p19q codel” gliomas as oligodendrogliomas, “IDH mutant 1p19q non codel” gliomas as enriched in low grade astrocytomas, and “IDH wt” gliomas as enriched in high grade astrocytomas and glioblastomas [29,30] (Figure 1). Note that such enrichment does not imply the exclusivity of a group for a given histological assessment.

We here highlighted CCR1, CCR5, CCR6, CCR10, CX3CR1, CXCR2, CXCR4, ACKR1, ACKR2 and ACKR3 based on their average mRNA expression within at least one patient subgroup (threshold arbitrarily placed at average RSEM ≥ 4). For most of the selected receptors, mRNA expression increases with glioma grade. Note that we do not rule out that unselected receptors may yet be of interest. We will therefore focus this manuscript on these ten chemokine receptors and unravel relevant literature data to further discuss their respective contribution to glioma biology.

In the last two decades [31], extensive evidence has proven CXCR4 as significantly related to glioma malignancy [32,33,34]. Its crucial contribution to the disease is supported by the phase I/II clinical testing of CXCR4 inhibitors for GBM treatment (e.g., plerixafor) [35,36], as further discussed in Section 4.1. The clinical relevance of the other selected chemokine receptors is supported by more or less abundant (pre)clinical data from the literature, which mostly analyzed mRNA/protein expression in glioma tissue sample cohorts (vs control tissue samples) and related these to tumor grade and patient survival. Among the receptors that appear highly expressed in all gliomas, CX3CR1 is a macrophage associated receptor, whose expression has been shown similar in patient tissue from both low and high grade gliomas [37,38], substantiating the TCGA data in Figure 1. Of note, a specific CX3CR1 defective polymorphism (V249I) correlates with increased patient survival in patients with GBM [39] and LGG [40], stressing its important role in tumor maintenance. ACKR3, formerly known as CXCR7/RDC1, has also been investigated in glioma patient tissue where its expression pattern appears quite inconstant: several studies highlight an increased mRNA expression in GBM tissue samples compared to non-malignant brain samples [41,42], while other studies do not [43]. Moreover, TCGA data show CCR1 expression in glioma samples. Although this has not been exhaustively documented in patient tissue thus far, insights in CCR1 activity in glioma are currently emerging (see Section 4). In comparison to the above-cited receptors, CCR5, CCR6, CCR10 and CXCR2 all display moderate expression in glioma patients from TCGA database. Their expression in tumor tissue has been assessed in diverse studies and was found upregulated in glioma (compared to non-tumor samples), correlating with the tumor grade as well as with shorter disease-free and overall patient survival [44,45,46,47]. Higher expression of CCR5 and CXCR2 has also been associated with recurrent tumors [48,49]. The roles of ACKR1 and ACKR2 in tumor growth have also been evaluated in other cancer types [50,51] where their expression has been correlated with a reduced tumor growth and survival benefit (reviewed in [15,52]). However, the role of these atypical receptors in gliomagenesis remains to be elucidated.

## 4. Chemokine Receptors in Glioma Malignant Processes

GBM is an extremely heterogeneous tumor, endowed with high invasive capacity, harboring hypoxic areas, necrotic and proangiogenic environments. Such heterogeneity complicates GBM treatment and constitutes an immense challenge for neuro-oncologists. The Ivy Glioblastoma Atlas Project (IvyGAP) has addressed this intra- and inter-tumoral heterogeneity by correlating anatomo-histological features with gene expression data in a panel of GBM patients [26]. In this study, five separate areas were analyzed after laser microdissection: (1) leading edge (LE), outermost boundary of the tumor; (2) infiltrating tumor compartment (IT), intermediate zone; (3) cellular tumor (CT), core part of the tumor with high ratio of tumor cells vs. healthy cells; (4) pseudopalisading cells around necrosis (PAN), densely aligned tumor cells surrounding necrotic areas; (5) microvascular proliferation (MVP) marked by two or more blood vessels. This freely accessible anatomo-transcriptional atlas provides a valuable ground to interrogate gene function in GBM growth processes. Here, we utilize this IvyGAP resource to look at the expression of the selected chemokine receptors in these five regions of interest and further decipher their activity in these specific regions (Figure 2).

CXCR4 once again stands out as highly expressed in the pseudopalisading cells around necrosis (PAN) and microvascular proliferation (MVP) regions, respectively described as related to hypoxia and angiogenesis/immune regulation [26]. ACKR3 also appears associated with MVP regions. Additionally, these two receptors are detected in the central tumor (CT), infiltrative tumor (IT) and leading edge (LE) regions, where their contribution could be of variable nature (see Section 4.1, Section 4.2 and Section 4.3). CX3CR1 and CCR1 are also expressed and distributed in all tested regions. CCR5, CCR6, CCR10, CXCR2, ACKR1 and ACKR2 display moderate expression in GBM samples, regardless of the area, which is in line with the TCGA data from Figure 1.

Using a similar approach, another study described chemokine receptor profiling in different GBM subregions [53], which were isolated after 5-aminolevulenic acid (5-ALA) fluorescence-guided surgery of six newly diagnosed GBM patients [54]. GBM cells were isolated from the tumor core (strong fluorescence, ALA+), infiltrating area (pale fluorescence, ALA-PALE) and healthy tissue (no fluorescence, ALA−). CXCR4 and ACKR3 were found upregulated in tumor core GBM cells (without distinction of necrotic and/or angiogenic features) compared to infiltrating area and healthy tissue, which is supportive of the IvyGAP data. Conversely, CCR1 and CCR10 were found upregulated in GBM infiltrating area compared to the tumor core, which suggests a role for these receptors at the margin of the tumor, probably linked to cell invasion or communication with the tumor microenvironment (TME).

The different tumor subregions that were studied in this dataset were defined based on specific anatomopathological features, associated with important GBM-related mechanisms. In the following paragraphs, we will discuss the putative role of chemokine receptors in one or several key tumor processes that could be related to their expression in the aforementioned tumor areas.

### 4.1. Angiogenesis

Angiogenesis is an important feature of high-grade gliomas, supporting tumor cell survival and invasion [55]. In line with the IvyGAP analysis of chemokine receptor expression in MVP and PAN regions, studies have demonstrated that CXCR4 is largely expressed in endothelial cells of the normal brain, as well as in GBM blood vessels and hypoxic areas of necrosis [42]. CXCR4 is enriched in highly vascularized GBM tissue [56] and its role in hypoxia-induced angiogenesis has been widely documented [31,57]. CXCR4 inhibition using plerixafor was therefore proposed in combination with bevacizumab (anti-vascular endothelial growth factor monoclonal antibody) for diminishing resistance to this anti-angiogenic therapy and has thus far proven safe in patients with high-grade gliomas [36]. ACKR3 is also found in endothelial cells as well as in tumor cells and microglia in GBM patient tissue specimens [42,58]. Moreover, a role for ACKR3 in tumor neovascularization has been suggested using in vitro models of tube formation with glioma endothelial cells [59], breast cancer cells [60] or human umbilical vein endothelial cells [61]. In glioma cells, ACKR3 expression appears upregulated in hypoxic conditions [62]. Given the discernible expression of ACKR3 in MVP areas of GBM tumors, its contribution to the angiogenic mechanisms in glioma patients and its interplay with CXCR4 definitely warrant further investigation. Although less prominently expressed in the MVP region based on the IvyGAP data, CXCR2 has also been associated with neovascularization. It colocalizes with blood vessels in GBM patient tissue and functionally helps GBM cells to transdifferentiate and acquire an endothelial-like phenotype, inducing vascular mimicry [47]. Finally, a co-culture model of glioma cells with normal astrocytes suggests that astrocyte-mediated production of CCL20 facilitates CCR6-expressing GBM cell adaptation to hypoxic TME via upregulation of hypoxia-inducible factor 1-alpha (HIF1-α). In particular, xenografts lacking CCR6 showed an impaired vascularization and reduced adaptability to hypoxic stress, supporting a role for this axis in GBM [63].

### 4.2. GBM Cell Migration and Invasion

Although not extremely prominent, several of the selected receptors are expressed at the invasive front of the tumor, which may suggest their involvement in GBM cell incursion through the brain parenchyma. CXCR4 expression has been associated with the extent of tumor cell dissemination within the patient brain (based on tumor imaging features) [34]. Preclinical models of gliomas highlighted its role in mediating cell migration and invasion [64,65], especially in the migration of specific “stem-like” cell subsets (see Section 4.3). In contrast, the activity of ACKR3 in glioma cell motility remains elusive and its function in the invasion of other cancer cell types is still a matter of debate. Indeed, in head and neck squamous cell carcinoma patients, ACKR3 expression has been associated with increased lymph node metastasis rate [66] and the relationship between CXCR4 and ACKR3 has also been linked to increased breast cancer metastasis in experimental models [67]. Contrasting results rather propose that CXCR4 and ACKR3 have distinct roles. CXCR4 seems to enhance cell invasiveness, while ACKR3 appears to be mainly associated with decreased invasive properties as well as inhibition of metastasis. ACKR3 is also suggested to promote tumor growth by stimulating angiogenesis [68].

Experimental data indicate that CCR5 and CXCR2 are involved in glioma cell invasion through tridimensional environments, when induced by co-cultured human mesenchymal stem cells [48,69] or endothelial cells [70] that were shown to secrete key chemokines. Hence, these receptors may play a role in GBM cell invasion through brain tissue.

### 4.3. GBM “Stem” Cell Properties and Resistance to Treatment

GBM progenitor/initiating/stem cell phenotype characterizes the self-renewing and plastic cell population within the tumor that sustains tumor growth and promotes resistance to treatment. Hence, significant efforts have been undertaken to specifically target these glioma stem cells (GSCs) (reviewed in [71]). GSCs have been associated with specific “vascular niches” within tumors [72], but also have been shown to be enriched in the cellular tumor (based on the IvyGAP data) [73]. CXCR4 was detected in cells expressing stem cell-associated markers (e.g., SOX2, KLF4, OCT4, NANOG) in both primary and recurrent GBM patient tissue sections [74]. Additionally, CXCR4 expression was found in patient-derived GSC primary cultures in vitro, where the receptor was implicated in cell survival, self-renewal and invasion upon xenografting [75,76,77]. Specifically, we previously showed that after orthotopic implantation, GSCs migrated toward the subventricular zone in an oriented, CXCR4-mediated fashion [78], which was associated with GSC protection from radiation therapy [79]. In contrast, only a minor subset of stem-like cells were found positive for ACKR3 in GBM patient tissue [74] and less information is available from in vitro patient-derived models. A study using selective ACKR3 modulators emphasized the involvement of ACKR3 in GSC growth in vitro together with CXCR4, although this report revealed that GSC tumor formation in vivo was independent of CXCR4 or ACKR3 activity [80]. Aside from sustaining tumor initiation, GSCs were shown to determine GBM cell response to therapy and were particularly suggested as crucial for the resistance to temozolomide (TMZ) [81]. A recent study has demonstrated that CXCR2 expression increased in patient-derived GSCs (expressing CD133, another stem cell-associated marker) upon treatment with TMZ in vitro. Activation of CXCR2-related pathways was indeed associated with alterations in the epigenomic landscape of cells, which impact GBM cell plasticity and resistance to TMZ [82]. Furthermore, CCR5 has been linked to TMZ resistance. Pericytes secrete CCL5 which activates CCR5 and downstream pathways in GBM cells. This leads to the activation of DNA damage response and thus reduces the efficiency of TMZ in killing GBM cells [83].

## 5. Chemokine Receptors in Diverse GBM Cell Subtypes

As mentioned above, the intratumoral heterogeneity of gliomas is largely accountable for therapeutic failure. Over the last years, the emergence of single-cell profiling technologies has deepened our understanding of glioma biology and the tumor heterogeneity that outreaches the anatomical level. Single-cell RNA sequencing is an advanced tool to decrypt the individual role of the different cell types forming the tumor, it allows to investigate the glioma heterogeneity at single-cell resolution. Several recent studies have shed light on the diverse malignant and non-malignant cell types that together compose gliomas and dictate their development, maintenance and response to therapy [6,7,27,84,85]. In high-grade gliomas, over a third of the tumor mass is constituted of non-malignant cells. The TME includes neuronal and glial cells, macrophage/microglial cells, representing the major immune cells component, endothelial cells and a low number of T cells [86].

Within the tumor, malignant/neoplastic cells were distinguished from non-malignant TME cell types using inferred copy-number alterations, and specific cell clusters were further categorized into TME subtypes based on their gene expression profile (for more detailed information, please refer to the original publications [7,27]). In the process of deciphering chemokine receptor function in gliomas, we explored two publicly available single-cell RNAseq datasets obtained from glioma patient tissue [7,27]. We looked into the expression of CCR1, CCR5, CCR6, CCR10, CX3CR1, CXCR2, CXCR4, ACKR1, ACKR2 and ACKR3 in the various cell type-related signatures that were reported (Figure 3).

In both datasets, CCR1 and CX3CR1 expression strongly correlated with the “macrophage” and “myeloid cell” signatures, while the two receptors were virtually absent from other cell types, unsurprisingly pointing to their key role in immune cell recruitment and function in gliomas. A high expression of CCR5 and moderate expression of CXCR2 is found in the same groups. CCR5 expression is also detected in the minor population of “T-cells”, as well as CCR6. CXCR4 is abundantly present in cells assigned to the “macrophage” and “myeloid cell” signatures, as well as in “vascular cells” and “neoplastic cells”, which is in line with the various roles of this receptor in diverse tumor-related processes. ACKR3 is detected in “neoplastic cells” and “vascular cells”, again supporting the data obtained from GBM patient tissue specimens [42,58]. Of note, this receptor is also abundantly expressed in “astrocytes” present in the tumor tissue. CCR10, ACKR1 and ACKR2 could be detected in different cell types at low level. The current knowledge on the function of these receptors in the respective cell entities will be further discussed in the following sections.

### 5.1. Tumor-Associated Macrophages (TAMs)

We previously mentioned that glioma TME largely contributes to the tumor bulk and influences tumor cell maintenance and growth. Tumor-associated macrophages (TAMs) derive from bone marrow circulating monocytes or from resident microglial cells and affect glioma progression in diverse manners depending on their activation status, interaction with TME components, phenotype or location within the tumor (reviewed in [87]). Thus far, TAMs are generally considered as supportive of GBM growth. CX3CR1-mediated macrophage infiltration into gliomas has been confirmed in patient tissue [88]. In GBM and LGG patients carrying the defective CX3CR1 V249I polymorphism [39,40], such infiltration is reduced which is associated with better prognosis.

Recently, a study investigated the single-cell transcriptome of multi-sector biopsies from 13 glioma patients (with various WHO grades) [89]. These data were used to reconstruct a ligand–receptor interaction map describing the most relevant chemoattractant relationships existing between tumor cells and TAMs in glioma TME. Nine chemokine receptors were detected in the 13 tumors, including CCR5, CCR6, CX3CR1, CXCR2 and CXCR4. This study reported that glioma cells overexpress CX3CL1, which is responsible for the recruitment of CX3CR1-expressing microglia and macrophages. CCR5 and CXCR4 were found on TAMs as well.

### 5.2. Tumor-Infiltrating Lymphocytes (TILs) and Other Immune Cell Types

Generally, gliomas are recognized as “cold” tumors endowed with poor immune response, where glioma cells expressing diverse immune checkpoint molecules (e.g., PD-L1) that hamper immune cell activation. Moreover, tumor infiltrating lymphocytes (TILs) poorly penetrate tumors, among which regulatory lymphocytes (T_reg_) secrete immunosuppressive cytokines (IL10 and TGF-β) and cytotoxic T cells exhibit a specific exhaustion profile (expression of PD-1 and CTLA4). In addition to the extensive glioma heterogeneity, such immune suppressive environment makes glioma refractory to targeted immunotherapy [90]. Literature data suggest that the level of TILs varies between different glioma genomic subtypes, with high grade (IDHwt) gliomas showing the highest TIL amount and the worse prognosis [91,92]. This encourages to (1) consider genomic profiles for predicting response to immunotherapy and (2) better understand and modulate TIL function and access to the tumor. To that purpose, regulating chemokine receptor function is of interest. The aforementioned report on GBM single-cell transcriptome confirmed that TILs express CCR6 (corroborating the data from Figure 3), as well as CCR5 and CXCR4, which all could contribute to lymphocyte recruitment toward the tumor [89].

Other immune-related cell types such as neutrophils, dendritic cells, myeloid progenitors and hematopoietic stem cells could also be found in gliomas [86,93] and their roles in glioma development and response to therapy are still under investigation. The recent literature provides pieces of information regarding the activity of chemokine receptors in these subsets. Early studies of TME in mouse models allowed to identify immature and immune-suppressive myeloid cells within solid tumors, which were called myeloid-derived suppressor cells (MDSCs) (likely encompassing diverse cell entities). Although efforts are currently carried out to standardize nomenclature and characterization of these cells, the MDSC term is still often used. MDSCs isolated from glioma patient tissue could be classified in monocytic (M-MDSCs) and granulocytic subsets (G-MDSCs). G-MDSCs presented increased CXCR2 expression but showed minor accumulation in the tumors compared to M-MDSCs [94]. Accordingly, CXCR2 was associated with neutrophils in the aforementioned single cell mapping of glioma TME components [89]. Of note, the degree of neutrophil infiltration has been positively correlated with glioma severity [95,96].

Efforts still have to be carried out to decipher the functional aspects of the complex glioma-associated immune orchestra to eventually shed new light on effective treatment options, which could rely on the modulation of chemokine-mediated immune cell recruitment to the tumors.

### 5.3. Vascular Cells

Endothelial cells from brain capillaries, as well as contiguous pericytes and astrocytic feet, are key components of the blood-brain barrier (BBB), which constitutes a selective filter that tightly regulates brain penetration of a variety of molecules and compounds. The integrity of this BBB is compromised in brain tumors [97], and endothelial cells exhibit various molecular alterations that reflect on their dysfunction, anatomical location, and variable permeability [98]. The expression of chemokine receptors in endothelial cells from glioma tissue has been detailed in Section 4.1 together with their role in angiogenesis. The implication of CXCR4 and ACKR3 in this process has particularly been documented. However, CXCR4 and ACKR3 expression is not specific to glioma-associated endothelial cells, and both receptors are also detected in endothelial cells from the developing brain [99] or from the adult brain [100]. A recent study developed an ACKR3 knock-in mouse model and highlighted ACKR3 expression in the cerebral vasculature, distributed across various brain structures [101], thus stressing a role of this atypical chemokine receptor in brain physiology that deserves deeper investigation.

Pericytes also play a pivotal role in the BBB maintenance. In GBM, pericytes exhibit specific genetic alterations. They mostly derive from GBM stem cells, which are recruited to blood vessels via CXCL12/CXCR4-mediated axis and evolve toward pericytes that contribute to vascular niche remodeling [102] and modulate GBM cell activity. Pericytes secrete CCL5, which binds the CCR5 receptor expressed by GBM cells. This interaction triggers the activation the DNA damage response, thereby overcoming TMZ-induced cell death. Inhibiting CCL5/CCR5 signaling abrogates the protective effects of pericytes against GBM and improves the efficacy of TMZ [83].

### 5.4. Non-Malignant Glial Cells and Neurons

As shown in Figure 3, ACKR3 as well as CXCR4 appear to be expressed also in non-malignant brain cells, notably in astrocytes. A study previously reported the presence of ACKR3 in adult rat astrocytes and further showed that its expression increases upon non-cancerous, neuroinflammatory conditions. ACKR3 is also detected in human astrocytes from the brain cortex and hippocampus and in oligodendrocytes and oligodendrocyte precursor cells (OPCs) [103]. In addition, preclinical models have shown the physiological role of ACKR3 in adult neuron physiology [104] and during development [105,106]. Overall, aside from the expression of ACKR3 and CXCR4 in glioma cells as well as in multiple cell subtypes from the TME, it appears that cell components of the neighboring healthy brain tissue require ACKR3 and CXCR4 for their maintenance and function, which could complexify their targeting in GBM therapy. Similarly, CCR10 and ACKR2 were also found to be expressed in astrocytes, albeit at lower level than ACKR3, while a small population of oligodendrocytes and OPCs express CCR6, CCR10 and ACKR2.

## 6. Conclusions

Gliomas are tumors of the central nervous system that remain associated with dismal prognosis in spite of innovative diagnostic strategies and modern therapies. Chemokines and their receptors play crucial roles in glioma development and progression and therefore constitute attractive candidates for targeted treatment. However, although their implication and targeting in in vitro and in vivo rodent models are well documented, especially for CXCR4 and ACKR3, their clinical relevance requires confirmation with patient data and biological material. Further analysis in terms of brain region distribution and by cell type enrichment is also necessary to better understand the complex roles of chemokine receptors within the highly diversified glioma landscape.

We found an overall good coverage and concordance of the different datasets used for the present analysis and congruence with targeted reports from the literature describing patient-derived material (Table 2). In this study, we focused on CCR1, CCR5, CCR6, CCR10, CX3CR1, CXCR2, CXCR4, ACKR1, ACKR2 and ACKR3, whose expression is detected in patient glioma tissue and rather well correlated with disease severity. With the aim of shedding light on chemokine receptor function in glioma physiopathology, this analysis integrated data from (1) publicly available bulk and single-cell transcriptomic datasets providing various types of information together with (2) evidence from the literature.

CXCR4 emerged as the most prominent receptor expressed on many different cell types within the tumor and associated with various tumorigenic processes such as angiogenesis, cancer cell invasion and resistance to treatment. This review also highlights ACKR3 as a multifaceted player in almost every of these glioma-related cellular processes and subtypes and prompts researchers in the field to further apprehend the subtleties of the CXCR4/ACKR3/CXCL12 triad in glioma. The importance of the CXCL12/CXCR4//ACKR3 axis in GBM is emphasized by multiple efforts toward the clinical translation of related inhibitors largely validated at the preclinical level [32,36,107,108,109]. A phase I/II clinical trial (NCT04121455) is currently investigating the impact of the CXCL12 inhibitor olaptesed pegol or NOX-A12 as part of combination therapy with radiation therapy and bevacizumab. Another phase I/II trial (NCT01977677), aiming at studying the safety and efficacy of the CXCR4 inhibitor plerixafor, after chemo/radiotherapy with the chemotherapeutic agent temozolomide (TMZ), suggests CXCR4-targeting as beneficial for patient survival and local control of tumor recurrence [35]. Finally, a clinical study (NCT03746080) has been recently initiated to better characterize the use of plerixafor in combination with whole-brain radiation therapy and TMZ.

Other receptors such as CCR5, CCR6, CXCR2 and CX3CR1 were mostly identified for their expression and function in immune cells from the tumor microenvironment. Despite the lack of supporting data from the literature, CCR1, CCR10, ACKR1 and ACKR2 also appear as significantly expressed in glioma tissue and deserve thus deeper investigation.

Although our study focuses on human classical and atypical chemokine receptors, Herpesviridae-encoded G protein-coupled receptors (GPCRs), homologous to human chemokine receptors, were also proposed to be important players in GBM. For instance, HCMV encodes for four viral GPCRs (US27, US28, UL33 and UL78) among which the oncomodulatory activities of US28 and UL33 have been recently described in GBM models [110,111,112].

Overall, this review highlights the intricacies of chemokine receptor activity in glioma, from central roles in glioma cells to key functions in TME partners and tumor-associated vasculature. It also highlights the complex and sometimes opposing roles certain receptors may have in GBM and related TME, making their targeting challenging and the benefits thereof uncertain. Therefore, the present study constitutes a valuable tool to gain awareness on receptor expression and function in GBM, which is fundamental for the development of efficient therapeutic approaches that would have the chemokines-chemokine receptors axis as main target.

## Figures and Tables

**Figure 1 cancers-14-00130-f001:**
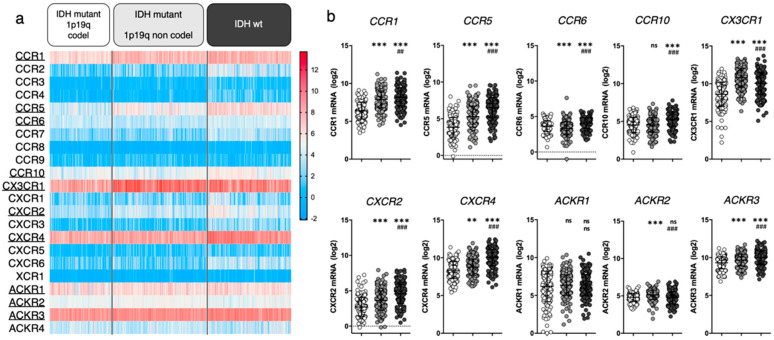
Chemokine receptor expression in glioma patients (TCGA LGG-GBM dataset [25], GlioVis platform^25^). (**a**) Heatmap displaying log2 RSEM value for the 22 receptors of interest. Each cell represents one patient. The receptors that will be highlighted in this review are underlined. (**b**) For every receptor, log2 RSEM values were grouped into 3 categories (based on glioma genomic features). Each dot represents one patient. Data are downloaded from http://gliovis.bioinfo.cnio.es (accessed on 23 November 2021), data are represented as Mean ± SD, and analyzed via one-way ANOVA (vs. “IDHmut 1p19q codel”: ** *p* < 0.01; *** *p* < 0.001 and vs. “IDHmut 1p19q non codel”: ## *p* < 0.01; ### *p* < 0.001).

**Figure 2 cancers-14-00130-f002:**
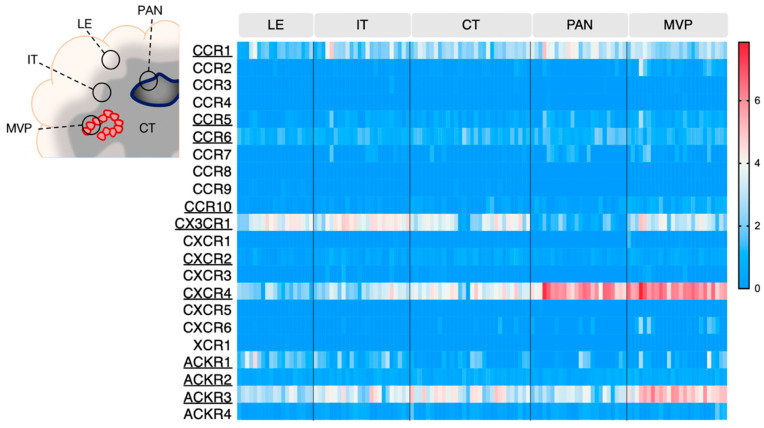
Chemokine receptor expression in various areas of GBM tumors (IvyGAP project). The heatmap displays log2 RSEM value for each receptor, in the various tumor subregions. Each cell represents one sample. Legend: LE: leading edge; IT: infiltrative tumor; CT: cellular tumor; PAN: pseudopalisading cells around necrosis: MVP: microvascular proliferation. Data is downloaded from https://glioblastoma.alleninstitute.org (accessed on 23 November 2021).

**Figure 3 cancers-14-00130-f003:**
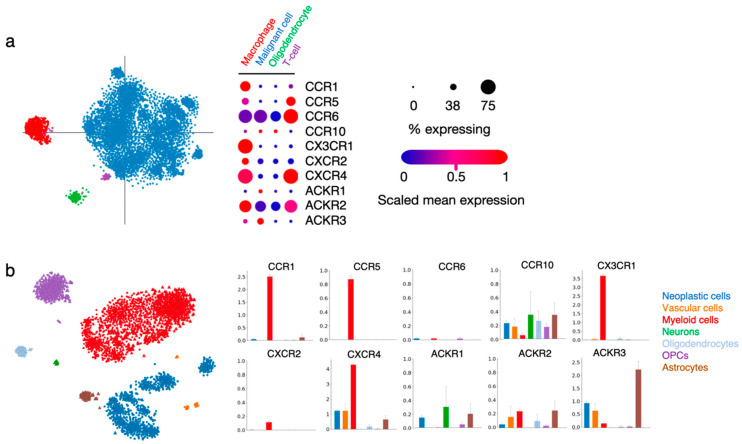
Chemokine receptor expression in various cell types within patient glioma samples. (**a**) Single cell RNAseq data from Neftel et al. (2019, Cell) [7] show the expression of the ten selected receptors in cells regrouped in four specific annotations (macrophage, malignant cell, oligodendrocyte and T-cell). The “% expressing” value indicates the proportion of cells in the signature that are positive for a given transcript, and the “scaled mean expression” is relative to each gene’s expression level (logTPM) across all cells within the signature (https://singlecell.broadinstitute.org/single_cell/study/SCP393/ (accessed on 23 November 2021). (**b**) Single-cell RNAseq data from Darmanis et al. (2017, Cell Reports) [27] show the expression of the ten receptors of interest in cells regrouped in seven specific annotations (neoplastic cells, vascular cells, myeloid cells, neurons, oligodendrocytes, oligodendrocyte precursor cells (OPCs) and astrocytes). Bar plots indicate log2CPM values (http://www.gbmseq.org/ (accessed on 23 November 2021)).

**Table 1 cancers-14-00130-t001:** General information about the datasets used in the review.

	1	2	3	4
Publication, project	[25] The Cancer Genome Atlas (TCGA) Project	[26] Ivy Glioblastoma Atlas Project	[27]	[7]
Selected information	Gene expression in three glioma subgroups (correlated with severity)	Gene expression in five anatomical locations within GBM tumors	Gene expression in 4 different glioma-related cell subtypes	Gene expression in 7 different glioma-related cell subtypes
Online tool	http://gliovis.bioinfo.cnio.es [28] (accessed on 23 November 2021).	https://glioblastoma.alleninstitute.org (accessed on 23 November 2021).	http://gbmseq.org/ (accessed on 23 November 2021).	https://singlecell.broadinstitute.org/single_cell/study/SCP393/ (accessed on 23 November 2021).
Method	Bulk RNAseq (HiSeq)	Bulk RNAseq (HiSeq) after laser microdissection	Single cell RNAseq (NextSeq)	Single-cell RNAseq (SMART-Seq2)
Datasets, number of samples and patients	Brain lower grade glioma, LGG (513 patients) Glioblastoma, GBM 154 patients)	Glioblastoma (122 samples/10 patients)	Glioblastoma (3589 cells/4 patients)	Adult and pediatric glioblastoma (IDHwt) (7930 cells/28 patients)
Data expressed as	Log2 RSEM	Log2 RSEM	Log2 CPM	Log TPM

Legend: CPM: counts per million; IDHwt: IDH wild-type; RNAseq: RNA sequencing; RSEM: RNA-Seq by expectation maximization; TPM: transcripts per million.

**Table 2 cancers-14-00130-t002:** Summary of chemokine receptor function in GBM tumorigenic processes and in GBM cell subtypes. Legend: TAMs: tumor-associated macrophages; TILs: tumor-infiltrating lymphocytes; **IV**: evidence from in vitro experiments (GSCs and others); **P**: evidence from patient tissue; **?**: still debated.

Role in		*CCR1*	*CCR5*	*CCR6*	*CCR10*	*CX3CR1*	*CXCR2*	*CXCR4*	*ACKR1*	*ACKR2*	*ACKR3*
Processes	Angiogenesis			** IV **			** P **	** P **			** P **
	Invasion		** IV **				** IV **	** P **			** ? **
	Stem cell properties						** IV **	** P **			** ? **
	Resistance		** IV **				** IV **	** P **			
Cell types	Tumor cells							** P **			** P **
	Vascular cells							** P **			** P **
	TAMs/Microglia		** P **			** P **		** P **			
	TILs		** P **	** P **				** P **			
	Neutrophils						** P **				
	Normal glial cells							** P **			** P **

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
