# Peer review of "Patient-Oriented Perspective on Chemokine Receptor Expression and Function in Glioma"

_cancers, 2021, doi:10.3390/cancers14010130_

Round 1

Reviewer 1 Report

Dear Authors,

First of all I enjoyed to read your manuscript- it is of great interest to researcher in that field!

But I have some criticism:

  1. When writing a scientific article in English it's always better to avoid long sentences (more than 40-50 words), especially if it´s possible to avoid them.
  2. Methodology: I think it is not well detailed and explained (it took me too much time to understand ...).
  3. Conclusion: This is the overall main point that we want the reader to remember. However in this conclusion we can notice a hard table (Table 2) to decipher. I suggest modifying that table and making it clearer, especially the aim of tables in articles is to make it simpler to understand and not the opposite.

Author Response

We sincerely thank the Reviewer 1 for considering our manuscript as a valuable addition to the field, and for providing suggestions for improvement.

  1. Given the 5-day delay before resubmission, we couldn’t send the manuscript for external English writing evaluation. Nonetheless, we carefully proofread the manuscript and modified too long sentences for an improved reading.
  2. We added more precise information on the methodology we used for establishing the analysis and the figures, in the text as well as in Table 1.
  3. Finally, we improved the clarity of Table 2, which initial format was not optimal, hence definitely not helping the reader to easily understand the conclusion.

Please find major modification highlighted in yellow through the text.

Reviewer 2 Report

Dear Authors,

Your review on the chemokine receptor expression and function in glioma is a very valuable contribution to the field of glioma research. The available data of the scientific community on these chemokine receptors is comprehensive, well analyzed and presented in a very clear way. The only data that could be presented in a more simple way is table 2. At least in my document the table is split over two pages and the words are somewhat squeezed in. I think a very simple presentation by only using different colors would be sufficient to carry the main message.

Best regards

Author Response

We sincerely thank the Reviewer 2 for supporting the rationale of our analysis and for considering our manuscript as a significant contribution to the glioma research field. Based on the Reviewer’s comment, we improved the clarity of Table 2, which initial format was not optimal, hence definitely not helping the reader to easily understand the conclusion. We hope the table is now more readable.

Reviewer 3 Report

This review is apporopiate for acceptance,but few correction are needed.

  1. GB has tow typr. Authors classification so not contain IDH mutant GB, so-called secondary GB. authous have to added the biological differenfe obetween de nova GB and secondary GB.
  2.  Neovasculature of GD consistent of endothelial cell and pericytes. authors have to add the findings of pericytes.
  3. Endothelial cells in neovasculature of GB is different character of  endothelial cells inf normal cerebral capillariesm, the discription of difference between EC in GB and EC in normal capillaries.

Author Response

We are thankful to Reviewer 3 for providing relevant suggestions. Please find in the attached file some more explanations and/or modifications that were implemented in the manuscript, following the reviewer’s advice.
